# Light Transmission Characteristics and Cytotoxicity within A Dental Composite Color Palette

**DOI:** 10.3390/ma16103773

**Published:** 2023-05-16

**Authors:** Nicoleta Ilie, Andrei Cristian Ionescu, Karin Christine Huth, Marioara Moldovan

**Affiliations:** 1Department of Conservative Dentistry and Periodontology, University Hospital, Ludwig-Maximilians University, Goethestrasse 70, D-80336 Munich, Germany; khuth@dent.med.uni-muenchen.de; 2Oral Microbiology and Biomaterials Laboratory, Department of Biomedical, Surgical and Dental Sciences, Università Degli Studi di Milano, Via Pascal, 36, 20133 Milano, Italy; andrei.ionescu@unimi.it; 3Institute of Chemistry Raluca Ripan, Babes-Bolyai University, 30 Fantanele St., 400294 Cluj-Napoca, Romania; marioara.moldovan@ubbcluj.ro

**Keywords:** light transmission, absorbance, cytotoxicity, shade, resin-based composite

## Abstract

Modern light-cured, resin-based composites are offered in a wide range of shades and translucencies. This large variation, created by varying the amount and type of pigmentation and opacifiers, is essential to enable an esthetic restoration in each patient situation, but may affect light transmission in the deeper layers during curing. We quantified optical parameters and their real-time variation during curing for a 13-shade composite palette of identical chemical composition and microstructure. Incident irradiance and real-time light transmission through 2 mm thick samples were recorded to calculate absorbance, transmittance, and the kinetic of transmitted irradiance. Data were supplemented by the characterization of cellular toxicity to human gingival fibroblasts up to 3 months. The study highlights a strong dependence of light transmission and its kinetic as a function of shade, with the largest changes occurring within the first second of exposure; the faster changes, the darker and more opaque the material. Transmission differences within progressively darker shades of a pigmentation type (hue) followed a hue-specific, non-linear relationship. Shades with similar transmittance but belonging to different hues were identified, while the corresponding kinetic was identical only up to a transmittance threshold. A slight drop in absorbance was registered with increasing wavelength. None of the shades were cytotoxic.

## 1. Introduction

Light-cured, resin-based composites (RBCs) are a modern, esthetic option for the seamless, imperceptible restoration of tooth defects. They are offered in a wide range of shades and translucencies to meet all patient needs in a clinical situation. While finding the right shade is fairly easy, translucency requires special attention as it is a major factor in preventing a restorative material from matching natural teeth [1,2]. Beyond esthetical considerations, translucency affects the depth of cure in direct-light RBCs [3], as well as the cure quality of underlying, light- or dual-cure luting RBCs, when using indirect restorative materials [4]. Poor polymerization quality in depth is unfortunately not immediately noticeable in a clinical situation, but later manifests itself in higher monomer elution [5], increased toxicity [6,7] and possible hypersensitivities, reduced mechanical properties [7], low degree of conversion [5] and discolorations.

When the light from a curing device is directed onto the surface of the RBC filling to be cured, one part is reflected, one part is absorbed and scattered, and only a small remainder is transmitted in deeper layers. How much light is transmitted through a material during light curing [8] is the result of the complex interplay of surface properties, microstructure, material composition, and restoration geometry [9]. In this interaction, it has been shown that the reflected light at the outer surface of a dental ceramic or an RBC is appreciable, with a reflection of 60–75% related to the incident light [10]. For highly polished CAD/CAM RBCs, which must be translucent to allow for setting of an underlying luting material, these values range from 12.6% to 18.5% [11] and 11% to 27% for defined translucent CAD/CAM RBCs [12]. These results are sobering and show an obvious inefficiency of light curing [10], considering that the goal of polymerization is to bring as much of the incident light as possible into the deeper layers of the material or to the surface of the luting material. How much light is reflected is directly dependent on the refractive index of the material, i.e., the chemical composition of the individual material components, and can be calculated theoretically using the Fresnel equation, taking into account the refractive index of the medium in which it is cured (air) and the material [12]. The size, distribution, geometry, and volume fraction of the fillers [13] then add their contribution on how light is reflected.

In contrast to reflection, absorption occurs when atoms or molecules of the RBC’s constituents, such as monomers [14], filler particles [15,16], photo-initiator molecules [17], dyes, and pigments [14,18,19] take up the energy of a photon of light. On the other hand, scattering occurs at the interface between two different media, e.g., an inorganic filler/organic matrix in the case of reinforcement particles or an air/organic matrix in the case of porosity voids [15]. While porosity voids are inevitable in an RBC, the scattering can be controlled by reducing the refractive index mismatch [20] between the organic matrix and the inorganic fillers, as well as the size and amount of the fillers [3]. Approaching the refractive index of the individual components, as well as increasing the filler size to reduce the filler/matrix interface, reduces the amount of scattering [20]. When such effects are measured in real time during the polymerization of an RBC, there is, in addition to the consumption of the photo initiator, a dynamic of light scattering that can either increase or decrease to a plateau during curing, due to changes in the refractive index of the organic matrix as it converts from monomer to polymer [21]. In addition to these remarks, scattering is highest when the filler diameter approaches approximately half the wavelength of the incident light, i.e., ~0.2–0.3 µm [3]. 

Because absorption and scatter are difficult to measure individually, they are often reported together as absorbance. Absorbance can be calculated based on either the amount of light reflected by a sample, or the amount of light transmitted through a sample. With RBCs, the latter method is regularly used [10,11,12]; measurement consists of shining the light from the curing unit through an RBC sample of a certain thickness and recording at the bottom of the sample how much light and also, if the detector type allows, what wavelengths were transmitted through the specimens to the detector. The absorbance determined by spectrometric analysis was identified as a possible criterion to estimate an RBC’s surface degradation during aging, while the linear absorption coefficient is generally maintained during aging [12].

Since microstructure and material composition, as described above, play a decisive role in the transmitted light, the study aims to determine and quantify the differences in absorbance/transmittance of a material where only the shading varies. The optical properties of 13 different shades of a single RBC composition were then complemented by cellular toxicity against human gingival fibroblasts in eluates collected up to 3 months.

The null hypotheses tested are that absorbance/transmittance and toxicity are similar for a material with identical filler and organic matrix microstructure and chemical composition, regardless of its shade (hue or value).

## 2. Materials and Methods

### 2.1. Materials

Thirteen hue (A, B, C) and value (1 to 4) variations (A1, LOT 031805; A3, LOT 051813; A3.5, LOT 121702; A4, LOT 051802; B1, LOT 011701; B2, LOT 101701; B3, LOT 011701; C2, LOT 011701; C3, LOT 011701; A2O, LOT 101702; A3O, LOT 101702; BW (Bleach White, London, UK), LOT 081702; Inc (Schneide, Andover, MA, USA), LOT 051802) of the RBC Beautifil II LS (Shofu Inc., Kyoto, Japan) were analyzed. The material selected consisted of a monomer mixture of UDMA (urethane dimethacrylate), Bis-MPEPP (bisphenol A polyethoxy methacrylate), Bis-GMA (bisphenol A glycol dimethacrylate) and TEGDMA (triethylene glycol dimethacrylate). The 83% by weight and 69% by volume filler system was a blend of boroaluminosilicate glass, surface pre-reacted glass ionomer (S-PRG), and pre-polymerized fillers.

Real-time light transmission for up to 60 s of irradiation through 2 mm thick RBC samples was recorded with a spectrophotometer. The same device allowed characterization of the incident irradiance, using a violet-blue LED (Light-Emitting Diode) LCU (Light Curing Unit) (Bluephase^®^ Style, Ivoclar Vivadent, Schaan, Liechtenstein) for polymerization. These parameters were then used to calculate the absorbance and transmittance of the 13-shade palette, as well as the kinetic of transmitted irradiance. Data were supplemented by characterization of cellular toxicity to human gingival fibroblasts in eluates collected up to 3 months.

### 2.2. Methods

#### 2.2.1. Spectrophotometry: Incident and Transmitted Irradiance

Incident and transmitted irradiance through the analyzed RBCs were assessed on a laboratory-grade USB4000 spectrometer (MARC (Managing Accurate Resin Curing) System, Blue light Analytics Inc., Halifax, NS, Canada) referenced by the National Institute of Standards and Technology (NIST). The spectrometer employs a 3648-element Toshiba linear Charge-coupled Device (CCD) array detector and high-speed electronics (Ocean optic, Largo, FL, USA) and was calibrated using an Ocean Optics’ NIST-traceable light source (300–1050 nm). The system uses a CC3-UV Cosine Corrector (Ocean optic, Largo, FL, USA) to collect radiation over a 180° field of view, thus mitigating the effects of optical interference associated with light collection sampling geometry.

The incident irradiance (irradiance hitting the sample surface) was determined on five occasions by placing the LCU directly centered and perpendicular to the spectrophotometer sensor. In addition, light transmittance was measured in real time during the polymerization of the RBC, which was placed in cylindrical Teflon molds (6 mm diameter, 2 mm increment thickness, *n* = 3) directly over the spectrophotometer sensor. The measurement began when the LCU was switched on and was tracked over 60 s of exposure to light. For this purpose, the LCU was positioned with a mechanical arm directly above, perpendicular and centered on the sample surface. While the samples were polymerizing, the spectrophotometer measured the irradiance at the bottom of the samples in real time. The cylindrical molds containing the material were positioned centered on the spectrometer’s circular detector, which had a diameter of 3.9 mm. Consequently, the irradiance reaching this area was taken into account. Irradiances in a wavelength range of 360–540 nm were recorded individually at a rate of 16 recordings/s. The sensor was triggered at 20 mW.

Incident and transmitted light were then used to calculate transmittance and absorbance for the individual shade variations. In this context, transmittance (T) is defined as the ratio of transmitted (I_t_) irradiance (radiant power) to incident irradiance (I_0_): T = I_t_/I_0_(1)
where I_t_ is the irradiance after the beam of light passes through the specimen and I_0_ is the irradiance of the incident light hitting the specimen surface. Further, transmittance is related to absorbance by the expression: Absorbance (A) = − log(T) = −log(I_t_/I_0_)(2)

Defined as ratios of irradiance, transmittance and absorbance are dimensionless.

#### 2.2.2. Cytotoxicity to Human Gingival Fibroblasts

Cytotoxicity was assessed following the recommendation of ISO 10993-12 [22] for dental materials. Therefore, 5 mm × 2 mm cylindrical RBC specimens were prepared in previously sterilized molds, by pressing the unpolymerized material between two polyacetate strips. Polymerization was performed from above only, to simulate a clinical application and by placing the curing unit mentioned above, directly centered and perpendicular to the sample surface. An exposure time of 20 s was used for all 13 material groups, as recommended by the manufacturer. Samples and cell culture medium were incubated in 15 mL conical tubes (Falcon, Becton, Dickinson and Company, Franklin Lakes, NJ, USA) at a sample surface area/mL cell culture medium of 117.8 mm^2^. For each material group, tests were performed in triplicates; each triplicate was tested in four individual measurements. A Dulbecco’s Modified Eagle’s high glucose Medium (Sigma-Aldrich Co., St. Louis, MO, USA) supplemented with 10% fetal bovine serum (FBS) and 1% penicillin/streptomycin (PenStrepFa, Sigma-Aldrich Co.) was used as cell culture medium. Media without test specimens served as negative control and were collected in the same way as the eluates. Collected eluates were substituted with 3 mL of fresh culture medium for further storage. The collected eluates were frozen at −20 °C prior to testing. 

The employed colorimetric cell proliferation assay WST-1 (4-[3-(4-iodophenyl)-2-(4-nitro-phenyl)-2H-5-tetrazolio]-1,3-benzenesulfonate) test uses tetrazolium salts (Sigma-Aldrich Co.), which are cleaved by cellular mitochondrial dehydrogenases to form formazan, to detect cell viability. A direct correlation between the number of metabolically active cells and the amount of formazan dye produced has been demonstrated. The absorbance of the dye was measured spectrophotometrically [23].

Human gingival fibroblasts (HGF-1, ATCC^®^ CRL-2014™, American Type Culture Collection, Manassas, VA, USA) were cultured in the medium described above using sterile cell culture dishes (nominal size of 100/20 mm; CellStar^®^, Greiner Bio-One International GmbH, Kremsmünster, Austria) and then incubated in a humidified CO_2_ incubator (HERACELL 150i, Thermo Scientific, Waltham, MA, USA) at 37 °C, 5% CO_2_ and a 95% air atmosphere. The medium was changed 3 times a week and the cells were monitored by an inverted phase-contrast microscope (Axiovert 40 C, Carl Zeiss AG, Jena, Germany). The cells were rinsed with Dulbecco’s Phosphate Buffered Saline (Sigma-Aldrich Co.) after reaching 80% confluency and detached with a trypsin-EDTA solution (0.25% trypsin, 0.53% mM EDTA). Cell counting was performed using a mixture of 10 μL of cell suspension and 10 μL of trypan blue solution (T8154, Sigma-Aldrich Co.) using a counting chamber (Neubauer Improved Haemocytometer, Paul Marienfeld GmbH & Co. KG, Lauda-Königshofen, Germany). Cells at passage 10 were used for all assays. The cells obtained in this way were seeded into 96-well cell culture plates (Cat. No. 655 160, CellStar^®^, Greiner Bio-One International GmbH, Kremsmünster, Austria) in 100 μL of cell culture medium at a density of 5000 cells per well. The cell culture medium was replaced 24 h later by eluates and control media followed by a 24 h incubation period in the humidified CO_2_ incubator at 37 °C. Then, ten μL of cell proliferation reagent WST-1 were added to each well, and the plates were incubated for an additional two hours in the CO_2_ incubator. Absorbance was measured according to the manufacturer’s standard protocol at 440 nm and 600 nm [23], using a scanning multiwell spectrophotometer (Varioskan LUX Multimode Microplate Reader, Thermo Scientific, Waltham, MA, USA) and SKanIT RE for Varioskan (Ver.2.2, Thermo Scientific) computer software. Wells without cells served as blank control. Absorbance at 600 nm (A600) was subtracted from that obtained at 440 nm (A440) to account for background variation of the plate. Cell viability is expressed as a percentage of cell viability compared to the negative control (untreated cells) using the following equation:(3)$$\%\;\mathrm{Viability}=100\times \left(A440-\left.A600\right)\right.\;\mathrm{treated}\;/\;\left(A440-\left.A600\right)\;\mathrm{control}\right.$$

The positive control of the test was a solution of 10% ethanol in cell culture medium. The results were interpreted in accordance with ISO 10993-5 [24].

### 2.3. Statistical Analyses

A Shapiro–Wilk test indicates a normal distribution of the variables, allowing for a parametric approach. One- and multiple-way analysis of variance (ANOVA) and Tukey’s honestly significant difference (HSD) post hoc-test at an alpha risk set at 5% were used. (SPSS Inc., Version 28.0, Chicago, IL, USA). 

## 3. Results

### 3.1. Incident Irradiance

The LCU used for polymerization is a violet-blue LED with a peak at 408 nm in the violet wavelength range and one at 458 nm in the blue wavelength range. The spectral distribution of the LCU is presented in Figure 1a. The irradiance of the LCU over the full wavelength range when placed directly on the sensor was (1391.3 ± 5.8) mW/cm^2^ and represents the incident irradiance for the RBC samples. When the LCU is started, the maximum irradiance is reached within 0.3 s and maintained over the entire exposure time. The short drop at 30 s represents the end of the exposure program and the restart of the LCU to meet the 60 s target (Figure 1b).

### 3.2. Light Transmission, Kinetic of Light Transmission, and Absorbance

Figure 2 summarizes the real-time measured transmitted irradiance at the bottom of the 2 mm samples during a 60 s light exposure. In all groups, the transmitted irradiance increased with exposure time and reached a plateau after 15 s exposure time at the latest. Similar to Figure 1b, the drop in the curves at an exposure time of 30 s represents the end of the exposure program, which had to be restarted in order to achieve the target exposure time of 60 s. A sharp drop in transmitted irradiance was observed relative to the incident irradiance of (1391.3 ± 5.8) mW/cm^2^. Values varied from 50.9 mW/cm^2^ for A4 to 226.3 mW/cm^2^ for Inc, thus representing no more than 4 to 16% of the irradiance reaching the sample surface. 

The data presented in Figure 2 were then used to calculate the rate of variation in transmitted irradiance during polymerization. The data were fitted using an exponential function and are shown in Figure 3a,b as examples for the shading A1 to A4 and B1 to B3. The results of the fit for all shades are summarized in Table 1. 

Percent transmittance is given in descending order in Figure 4. In this descending sequence, one-way ANOVA identified similar transmittance for A1 and B2, A3 and B3, A2O and C2, A3.5 and C3, and C3 and A3O. The *p*-values of these comparisons are given in the figure. The absorbance summarized in Figure 5 gives the opposite relationship, with the highest absorbance calculated for A4 and the lowest for the translucent material group Inc.

In addition to the absorbance calculated over the entire spectral range of the LCU, Figure 6 also shows the variation of the absorbance in detail per wavelength in the range of interest for the polymerization, which represents the area around the LCU peak in the blue wavelength range. The ranking of the materials relates to the absorbance presented in Figure 5. In all groups, a slight drop in absorbance is registered with increasing wavelength.

### 3.3. Cytotoxicity

Figure 7 shows the percent viability of HGF-1 cells exposed to eluates from the 13 different shades at different elution times, up to three months, relative to the corresponding negative control (NC). None of the analyzed shades indicated cytotoxicity at any of the elution times.

## 4. Discussion

The shade palette analyzed in the present study included pigment variations that labeled the material’s hue as “A” for reddish brown, “B” for reddish yellow, and “C” for gray, while the increasing annexed index (e.g., 1 through 4) indicated the value of the individual hue, with variations towards progressively darker shades. In addition, the palette contained two opaque dentine shades (A2O and A3O), one translucent enamel shade (Inc) and one very light, so-called bleach shade (BW). This wide range of shades was required to take into account the diversity of human teeth and to enable an esthetic, seamless restoration in every patient situation. However, an important aspect that cannot be neglected clinically is that the amount of light that penetrates a material in depth to enable proper polymerization depends crucially on the optical parameters. This was clearly reflected in the analyzed shade palette in a 5–6 times lower light transmission through 2 mm thick increments with shade A4 compared to Inc, a highly translucent shade intended for incisal restorations. To understand and quantify these differences, we selected shades of a single material to maintain the same chemical composition and microstructure, so that absorption and reflection of the individual components such as monomers [14], filler particles [15,16], and photo-initiator molecules [17] are similar, and differences between shades can simply be related to the type and amount of pigments and opacifiers [14,18,19]. An amendment needs to be made to this reasoning, as there was no exact information as to whether the amount of photo-initiator had been altered for very dark shades compared to lighter ones in order to enable more efficient polymerization, as a consequence of a small amount of light reaching the depth. In addition, since voids are also responsible for scattering in a material [15], their probability, either in the uncured paste or induced during the sample preparation, is assumed to be similar in all shades, since the viscosity/rheology or material handling are not altered by small variations in pigments.

Light attenuation was monitored in real time in the present study design during a 60-s exposure, a much longer time than is required for full polymerization of such materials, to ensure saturation occurred for all shades. An advantage of the method used for measuring light transmittance is the very accurate and rapid data recording at 16 readings per second, which allowed calculation of the rate of change in transmitted irradiance. This was used as a characteristic of the polymerization process as a function of the shading factor in the more critical material depth, since in a clinical situation the analyzed material has to be placed in increments no thicker than 2 mm. The data show very clearly in this regard that the greatest change in transmitted light occurs within the first second of exposure, ranging from 63.3% for BW to 99.5% for A4, related to the maximum transmitted irradiance after full exposure (60s) in each individual shade. Note that the first reading in Figure 2 after the start of light exposure relates to the transmitted irradiance through the yet uncured material and is summarized in Table 1 as Ir_min_. From this point on, the light attenuation curves for all shades followed an exponential increase within the first few seconds of polymerization, ending in a plateau after 15 s at the latest. The Ir_min_ data, i.e., the light transmittance through the uncured paste, showed a decrease within a hue (A, B, or C) with increasing darkness (increased value, e.g., Ir_min_ varied from 73.8 mW/cm^2^ for A1 to 30.9 mW/cm^2^ for A4) and opacity (e.g., Ir_min_ was 65.1 mW/cm^2^ for A3 and 39.8 mW/cm^2^ for A3O). Recording the transmitted light through the uncured shades allowed for an interesting observation, namely that Ir_min_ decreased less with the value for the hue B (30% difference between B1 and B3) than for the hue A (40% difference between A1 and A3). In addition, for hue B, the differences between lighter shades (B1 vs. B2) were much larger than for darker shades (B2 vs. B3), while for hue A the opposite was observed. This allows the conclusion that there is neither a linearity in the light transmission variation through increasingly darker shades within one hue, nor a generally valid relationship that could be transferred to all pigmentation types. This different behavior may be related to the higher absorption in blue light of the yellow pigmentation, as the hue B was reddish/yellow colored, compared to the reddish/brown pigmentation of shade A. This clearly emphasizes that there are different relationships to be assumed between darker and lighter materials of the same hue. In addition, Inc prove to be very translucent already in the uncured state (I_rmin_ = 154 mW/cm^2^), validating the manufacturer’s intention for it to be used as an enamel replacement. The exponential increase in transmitted irradiance with exposure time, seen in all shades, showed that in addition to consumption of the photo-initiator, the refractive index match between inorganic fillers and organic matrix was improved during conversion from monomer to polymer, progressively allowing more light to be transmitted and thus to reach deeper layers [21]. 

The rate of variation of the transmitted irradiance exemplified in Figure 2a,b and summarized in Table 1 was able to distinguish very sensitively between the curing behaviors of each individual shade. The data were well fitted by an exponential function, as evidenced by high R-squared values (Table 1). In addition, the data show (e.g., hue A) a statistically significant increase (for statistically significant differences, please consider the given standard error for fit parameters “a” and “b”) in both parameters “a” and “b” with increased darker shades within one hue, indicating that the maximal attainable transmission was reached faster and started with a higher rate in the darker shades. Interestingly, the calculated kinetic indicated statistically similar parameters of fit (a and b) in the groups identified with statistically similar transmittance, namely, in increasing absorbance sequence: A1 and B2; A3 and B3; A2O and C2; and A3.5 and C3, but not for the group C3 and A3O. In this latter case, the exponential fit identified variation statistically faster and at a higher initial rate (higher a and b parameters) in A3O compared to C3, demonstrating that even if the final transmittance is similar in both shades, the speed to reach this value is faster in the more opaque material. The fact that discrepancies in the statistical similarity in the kinetic of transmitted irradiance at similar absorbance occur for more opaque or darker shades, but not for the lighter ones, indicates a threshold for this behavior, which for the analyzed shades lies between the shades A3.5 and A3O, i.e., for an absorbance greater than 1.17 but less than 1.26, corresponding to transmittances of 5.5% to 6.7%.

Considering that light attenuation increases exponentially with material thickness, following the Lambert–Beer law [11], the chosen study design only allowed comparison with studies using the same sample geometry. The measured light attenuation of 72% in the more translucent shade Inc up to 93.6% in the darkest shade A4 corresponds to what was described in the literature for CAD/CAM-RBCs and ceramics that comprises a range from 59.9% to 94.9% [8]. This aspect is of particular importance in a clinical situation where, in larger restorations, the RBCs are layers in increments that are cured separately. As the amount of light transmitted through a 2 mm increment is small, each increment should be properly cured individually and one should not consider that a bottom increment would receive additional light to complete its polymerization while a top layer is light cured. This recommendation applies to all shades in a clinical situation, as the light reaching a lower increment during the curing of the top one varied from 50 mW/cm^2^ in A4 to 226 mW/cm^2^ in Inc, a range that is too low to compensate for any insufficient polymerization.

In a similar comparison, the light attenuation of the tooth structure measured under identical conditions and geometries was found to be 89.4% [8], which would correspond to the shade BW in the present study. Note that the measured tooth structure was dry to mimic a clinical situation by isolating a tooth during restoration, as the translucency of the wet tooth structure can increase due to the lower refractive index difference between water and tooth structure versus air and tooth structure [8]. Moreover, the comparison must be viewed in terms of the amount of light lost when passing through a material at a depth of 2 mm, and not from an esthetic point of view. Note that, compared to RBCs, absorption is low in both enamel and dentin, while scattering is strong in dentine and weak in enamel [25]. 

Even if the chemical composition of the analyzed shades was identical, the characterization of the cytotoxicity [26] of each individual shade is of great importance, as it can be related to an insufficient monomer conversion, involving residual monomers [27] as a result of a lower amount of light reaching the depth of darker or more opaque shades. Note that unlike the light transmission measurements, the samples for the cytotoxicity analysis were run under strict clinical curing conditions, such that all samples were exposed for 20 s and from one side only. Since aging can also produce degradation products that can have a cytotoxic effect [28,29], the toxicity of the materials was evaluated for up to 3 months. Based on a recognized standard for testing the cytotoxicity of dental materials (DIN EN ISO 10993-5 [24]), we selected a commercially available HGF-1 cell line, not only because it originates from gingival fibroblasts that are in direct contact with or very close to dental materials under clinical conditions [30], but also because it allows for standardization as it is independent of individual donor variation. As for the assay used to assess cell viability, the WST-1 test is well standardized and accepted. The limitation of the test is that in the case of detected toxicity, defined as a reduction in cell viability greater than 30% compared to the negative control used in the experiment, the exact cause of a detected toxicity cannot be determined (DIN EN ISO 10993-5 [24]). None of the 13 analyzed shades reduced cell viability at all tested elution times up to 3 months and thus proved to be non-cytotoxic under the curing conditions described in the experiment. 

## 5. Conclusions

The study highlights a strong dependence of light transmission and its kinetic as a function of shade during polymerization in materials with similar chemistry and microstructure, with light attenuation varying from 72% to 93.6%. The greatest change in transmitted light occurs within the first second of exposure. There was neither a linearity in the light transmission variation due to increasingly darker shades within a hue (A or B), nor was there a general relationship that could be transferred to all hues. A slight drop in absorbance with increasing wavelength was observed. Despite large differences in optical properties, none of the shades was cytotoxic up to 3 months.

## Figures and Tables

**Figure 1 materials-16-03773-f001:**
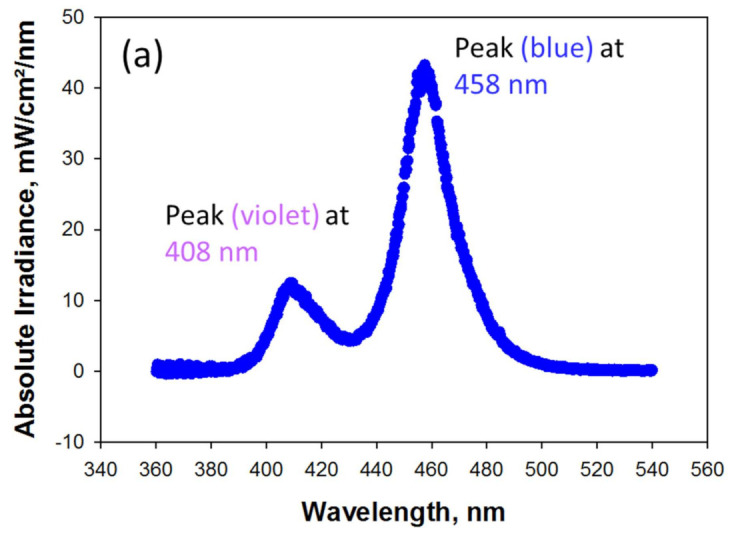
LCU characteristics: (**a**) spectral distribution; (**b**) variation of the irradiance within the 60 s exposure.

**Figure 2 materials-16-03773-f002:**
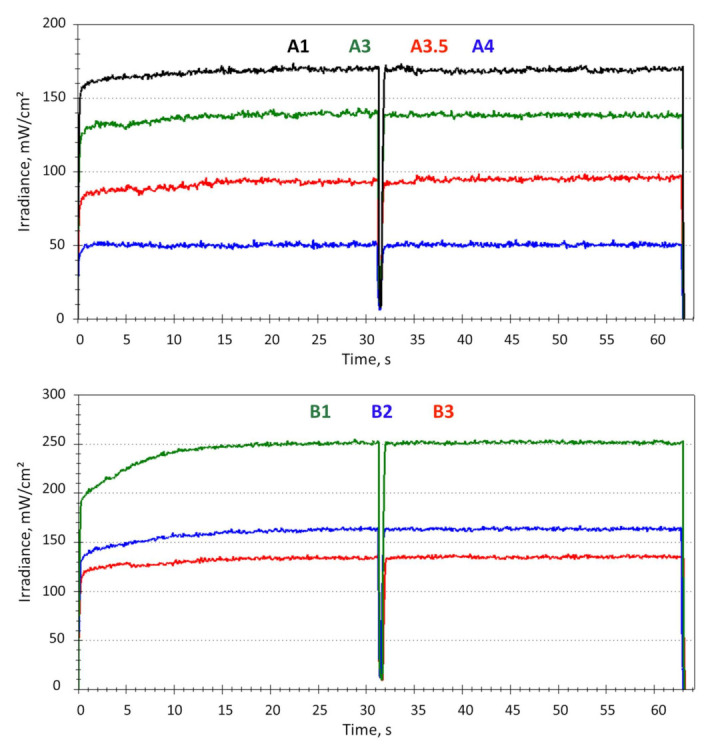
Variation of the transmitted irradiance through 2 mm thick samples of the 13 analyzed shade variations within 60 s exposure time.

**Figure 3 materials-16-03773-f003:**
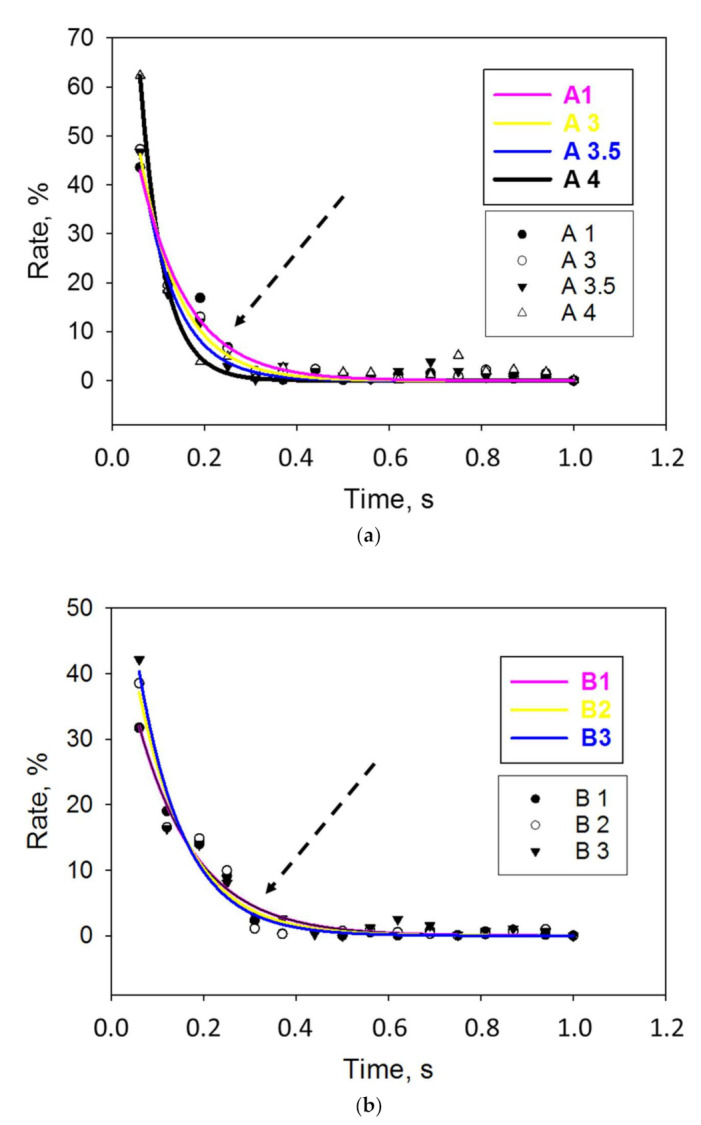
Rate of change in transmitted irradiance during polymerization within the first second of exposure exemplified for (**a**) shades A1 to A4 and (**b**) shades B1 to B3. The arrow marks the increasing value (1 to 4 and 1 to 3) within a hue.

**Figure 4 materials-16-03773-f004:**
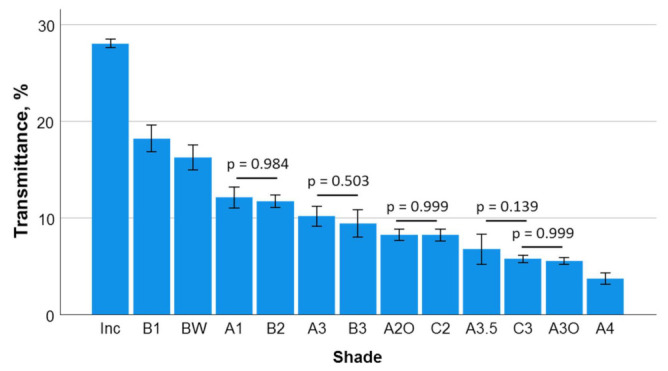
Transmittance (T) as a function of shade (mean values with 95% confidence interval). Data are presented in descending order of transmission. Statistically similar groups are denoted by a horizontal line while the *p*-value is displayed.

**Figure 5 materials-16-03773-f005:**
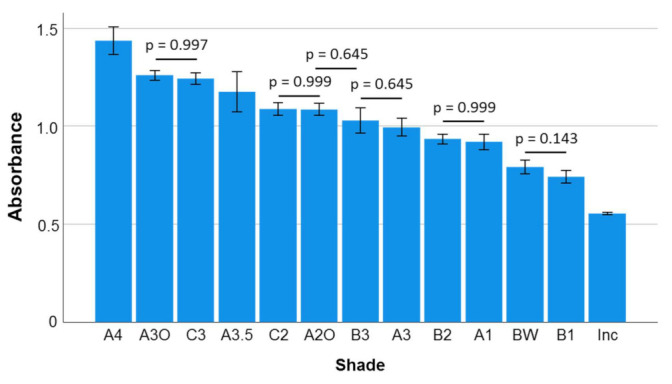
Absorbance (A) as a function of shade (mean values with 95% confidence interval). Data are presented in descending order of absorbance. Statistically similar groups are denoted by a horizontal line while the *p*-value is displayed.

**Figure 6 materials-16-03773-f006:**
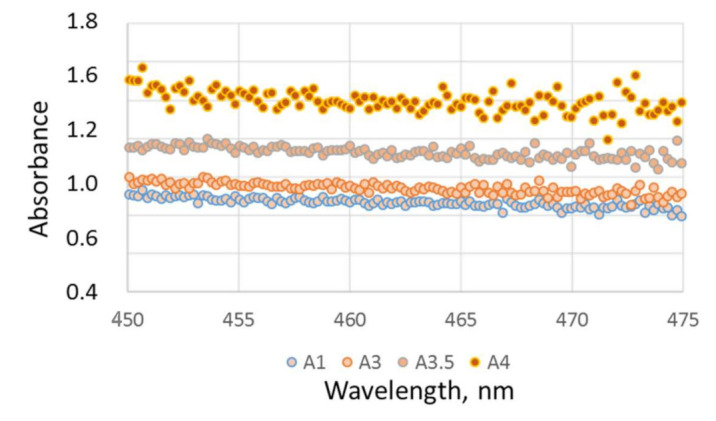
Absorbance (A) as a function of wavelength and shade.

**Figure 7 materials-16-03773-f007:**
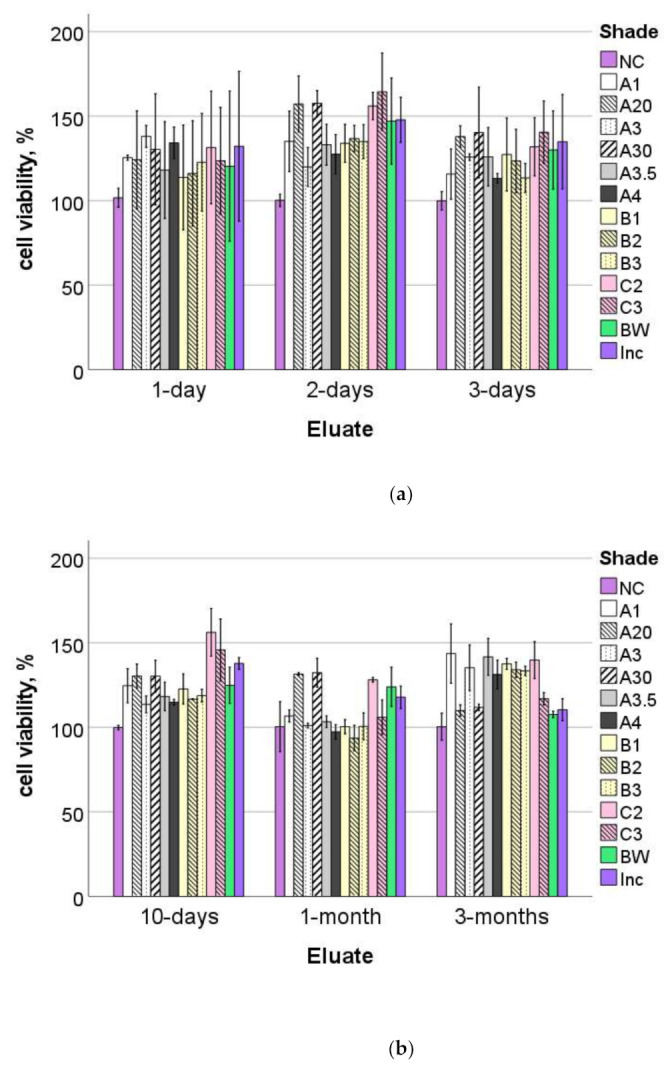
Cell viability (mean and standard deviation) in percentage of the negative control (NC), ordered by chronological elution times: (**a**) 1-, 2- and 3-day eluates and (**b**) 10-day, 1- and 3-month eluates.

**Table 1 materials-16-03773-t001:** Parameter of the exponential fitting curve (2 parameters, exponential decay) of the rate of irradiance with exposure time during the first second of exposure. R^2^ = fit of the model; a, b parameters of the exponential function with their standard error; % Ir after 1s = percentage of irradiation related to the 60 s exposure after the first second of irradiation; Ir_min_ = transmitted irradiance through the unpolymerized material at the first point of measurement; Ir_max_ = transmitted irradiance at 60 s irradiation.

(a) Shades A1–A4, A2O, and A3O							
Parameter	A1	A3	A3.5	A4	A2O	A3O	
R^2^	0.99	0.98	0.99	0.99	0.97	0.98	
a	76.30	92.38	101.87	203.61	77.02	148.44	
Std. Error	5.57	6.26	11.08	23.51	7.42	16.77	
b	9.60	11.55	13.27	19.76	11.29	17.68	
Std. Error	0.74	0.77	1.34	1.68	1.08	1.58	
% Ir after 1s	90.76	89.48	81.89	99.54	76.06	73.71	
Ir_min_, mW/cm^2^	73.80	65.13	45.08	30.93	47.52	39.84	
Ir_max_, mW/cm^2^	168.76	138.73	96.09	50.32	115.07	77.76	
**(b) Shades B, C, Inc, and BW**							
**Parameter**	**B1**	**B2**	**B3**	**C2**	**C3**	**Inc**	**BW**
R^2^	0.99	0.98	0.98	0.98	0.99	0.98	0.98
a	64.59	74.15	87.58	84.49	71.65	71.65	48.86
Std. Error	6.08	6.80	8.34	6.20	4.81	4.81	2.67
b	7.88	9.23	10.16	11.48	11.57	10.43	7.83
Std. Error	0.54	0.92	0.96	1.08	0.83	0.72	0.48
% Ir after 1s	69.41	75.95	79.53	79.15	84.13	70.26	63.30
Ir_min_, mW/cm^2^	80.09	62.44	56.26	52.06	34.63	154.06	69.57
Ir_max_, mW/cm^2^	254.65	163.03	137.00	115.56	80.05	390.75	231.08

## Data Availability

Data are available on request.

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
