# Peer review of "Light Transmission Characteristics and Cytotoxicity within A Dental Composite Color Palette"

_materials, 2023, doi:10.3390/ma16103773_

Round 1

Reviewer 1 Report

Congratulations on the manuscript. Interesting research idea and I believe it will contribute to the scientific community and researchers in the area.

Author Response

All comments to the corresponding author have been addressed independently below. The authors’ rebuttal is always in BLUE and where changes have been added to the revised manuscript in light of the reviewer's comments these are presented in RED.

The author would firstly like to thank the reviewers for taking the time to read and critically appraise the manuscript and secondly to thank the reviewers for their positive constructive comments in improving the work.

Comments and Suggestions for Authors

Reviewer comment: Congratulations on the manuscript. Interesting research idea and I believe it will contribute to the scientific community and researchers in the area.

Author’s response:  we are grateful for this appreciation. 

Reviewer 2 Report

This manuscript reviews the light transmission characteristics and cytotoxicity of dental composite for a 13-shade composite palette. The manuscript is well written and based on up-to-date literature and presents relevant findings from an original investigation. It is easy to read, and the paragraphs come from each other. Nevertheless, there are some issues to review before being suitable for publication:

Figure 1. Labels “a)” and “b)” are not properly fixed with the figure archive.

Figure 2. The figure quality of the image is poor. A better-quality image must be provided.

Figure 3. Review labels and be consistent with previously presented figures. E.g., “Time (s)” instead of “Time, s”.

Table 1. It should be presented in one table, not in two tables.

Figures 4 and 5. If the authors choose to display in the graph the p values of comparison between groups pairs, all comparisons must be presented. There is no intuitive way to know why some of the values are displayed and not others. Maybe displaying only statistically significant values of compared groups will be better.

For all the figures, review labeling, and set them consistent in the way you present data. E.g., “Time (s)” instead of “Time, s”.

Discussion. For this reviewer, a deeper discussion of composition regarding the nature of pigments or fillers must be provided since all the study design comes from the assumption that, quoting, “the chemical composition of the analyzed shades was identical”, but this assumption must be, at least, discussed.

Good enough.

Author Response

All comments to the corresponding author have been addressed independently below. The authors’ rebuttal is always in BLUE and where changes have been added to the revised manuscript in light of the reviewer's comments these are presented in RED.

The author would firstly like to thank the reviewers’ for taking the time to read and critically appraise the manuscript and secondly to thank the reviewers’ for their positive constructive comments in improving the work.

Comments and Suggestions for Authors

Reviewer comment: This manuscript reviews the light transmission characteristics and cytotoxicity of dental composite for a 13-shade composite palette. The manuscript is well written and based on up-to-date literature and presents relevant findings from an original investigation. It is easy to read, and the paragraphs come from each other.

Author’s response:  we are grateful for this appreciation.

Nevertheless, there are some issues to review before being suitable for publication:

Figure 1. Labels “a)” and “b)” are not properly fixed with the figure archive.

Author’s response:  thank you for the observation. We now placed the labels in the figures for more clarity.

Figure 2. The figure quality of the image is poor. A better-quality image must be provided.

Author’s response: Images have been revised, please take changes into account. Please note that the scatter in the images with low irradiances is measured and does not represent poor image quality.

Figur 3. Review labels and be consistent with previously presented figures. E.g., “Time (s)” instead of “Time, s”.

Author’s response:  Thank you for this relevant note. The error has been corrected to unify the style

Table 1. It should be presented in one table, not in two tables.

Author’s response:  Table 1 cannot be presented in one table as it will be very long or wide. The most economical representation using normal letter size was to split the 13 measured shades into two tables.

Figures 4 and 5. If the authors choose to display in the graph the p values of comparison between groups pairs, all comparisons must be presented. There is no intuitive way to know why some of the values are displayed and not others. Maybe displaying only statistically significant values of compared groups will be better.

Author’s response:  Groups are presented in descending order of measured properties. The p-value was only given for statistically similar groups marked by a vertical line. All other comparisons were statistically different and the p-value was lower than 0.001. Giving p<0.001 multiple times would load up the figures. It is a common way in science to mark statistically similar groups and back them up with the p-value. I've included a comment on this in the legend for clarity. Thank you for pointing out the incomprehensibility.

For all the figures, review labeling, and set them consistent in the way you present data. E.g., “Time (s)” instead of “Time, s”.

Author’s response:  This has been done, please note the changes above as well as changes in the manuscript.

Discussion. For this reviewer, a deeper discussion of composition regarding the nature of pigments or fillers must be provided since all the study design comes from the assumption that, quoting, “the chemical composition of the analyzed shades was identical”, but this assumption must be, at least, discussed.

Author’s response:  The similarity in the chemical composition of filler and matrix was the premise of the study design.  As mentioned under Materials and Methods, one composition was tested while the differences between shades related only to the pigments.  A chemical analysis was not performed in the present study. Since we worked with commercially available materials, as the reviewer knows, there is no information on the amount and type of pigments. A detailed discussion about the nature and amount of the pigments in the analyzed material cannot be conducted as this information is missing. Such effects can be analyzed only in experimental materials. However, we tried to address the effects we observed related to the different pigmentation. Please note the discussion we made regarding the differences between hues A and B and the interaction of potential yellow pigments with the blue light.

Reviewer 3 Report

The purpose of this study is to investigate the light transmission characteristics and cytotoxicity of a dental composite color palette. The manuscript has been well-written, but there are some concerns I would like to address as the following:

Major concern:

1.     What are the clinical impacts of this study? please add the discussion and suggestion that shows how the results of this study could be applied in clinical treatment.

2.     In a clinical situation, to restore the natural color of the teeth, multiple layers of composite resin will be applied. In these cases how the light transmission will be affected? This issue needs further discussion in the revised manuscript.

Minor concern:

1.     In the abstract, the results have been reported without any exact values, also statistical values such as p-values should be reported.

2.     Equations should be separated and numbered to make them easier to be followed.

3.     Plagiarism should be less than 15% (now it was 36%)

Author Response

All comments to the corresponding author have been addressed independently below. The authors’ rebuttal is always in BLUE and where changes have been added to the revised manuscript in light of the reviewer comments these are presented in RED.

The author would firstly like to thank the reviewers for taking the time to read and critically appraise the manuscript and secondly to thank the reviewers for their positive constructive comments in improving the work.

The purpose of this study is to investigate the light transmission characteristics and cytotoxicity of a dental composite color palette. The manuscript has been well-written, but there are some concerns I would like to address as the following:

 Author’s response:  we are grateful for this appreciation.

Major concern:

  1. What are the clinical impacts of this study? please add the discussion and suggestion that shows how the results of this study could be applied in clinical treatment.

Author’s response:  we extended the discussion with the suggestion made by the reviewer. Please consider the paragraph added to page 15.

  1. In a clinical situation, to restore the natural color of the teeth, multiple layers of composite resin will be applied. In these cases how the light transmission will be affected? This issue needs further discussion in the revised manuscript.

Author’s response:  Thank you for pointing this out. We completed the discussion by adding the clinical implication of the low amount of light transmitted through an increment. This aspect has been particularly addressed by the study design, as we deliberately analyzed all materials in 2mm increments, thus in the layer thickness considered clinically relevant.

Minor concern:

  1. In the abstract, the results have been reported without any exact values, also statistical values such as p-values should be reported.

Author’s response:  we understand the reviewer's concern, but an abstract is limited to 200 words and the study design is complex. Not all statistical evaluations can be mentioned and it would also be wrong to select a few. Please note the special features of the journal and its requirements, in which the abstract has a more informative character than regular dental journals, in which detailed statistical evaluations are requested.

  1. Equations should be separated and numbered to make them easier to be followed.

Author’s response:  We number the equations we used as suggested in the manuscript.

  1. Plagiarism should be less than 15% (now it was 36%)

Author’s response:  We were quite shocked about this affirmation and have checked the allegation of plagiarism with the editors. We have been confirmed that there is no plagiarism. Similarities in the description of the methods, equipment used, statistical evaluation methods, etc. are difficult to avoid and so it is very likely that some repetitions have occurred. We have done our best to minimize this aspect. This has nothing to do with plagiarism.

Reviewer 4 Report

The manuscript shows an interesting study to determine and quantify the differences in optical parameters such as absorbance/transmittance, and their real-time variation, during curing for a 13-shade composite palette of identical chemical composition and microstructure. Cellular toxicity assay against human gingival fibroblasts in eluates collected up to 3 months was also assessed.

The work is a very interesting topic that falls within the aims and scope of the journal, useful for exploring further possibilities in the field of application of modern light-cured, resin-based composites.

The paper is well written and organized, and the novelty of the paper is well explained in the introduction. It can be accepted for publication after addressing the following issues:

The abstract needs a revision about the number of words since it has to be a single paragraph of about 200 words maximum.

Figure2: please, provide for a more detailed figure caption for a clearer comprehension of each graph irradiance versus time

Figure 6. Normalize the figure layout. x and y titles need to be like the ones of the other figures

Reference 23. Accessed on?

Author Response

All comments to the corresponding author have been addressed independently below. The authors’ rebuttal is always in BLUE and where changes have been added to the revised manuscript in light of the reviewer's comments these are presented in RED.

The author would firstly like to thank the reviewers for taking the time to read and critically appraise the manuscript and secondly to thank the reviewers for their positive constructive comments in improving the work.

The manuscript shows an interesting study to determine and quantify the differences in optical parameters such as absorbance/transmittance, and their real-time variation, during curing for a 13-shade composite palette of identical chemical composition and microstructure. Cellular toxicity assay against human gingival fibroblasts in eluates collected up to 3 months was also assessed.

The work is a very interesting topic that falls within the aims and scope of the journal, useful for exploring further possibilities in the field of application of modern light-cured, resin-based composites.

The paper is well written and organized, and the novelty of the paper is well explained in the introduction.

Author’s response:  we are grateful for this appreciation.

It can be accepted for publication after addressing the following issues:

The abstract needs a revision about the number of words since it has to be a single paragraph of about 200 words maximum.

Author’s response:  Thank you for that pertinent observation - we have reduced the abstract to the journal request and restricted it to 200 words.

Figure2: please, provide for a more detailed figure caption for a clearer comprehension of each graph irradiance versus time

Author’s response:  Figs. 2 have been edited for more clarity.

Figure 6. Normalize the figure layout. x and y titles need to be like the ones of the other figures

Author’s response:  Figs. 6 have been edited, as suggested.

Reference 23. Accessed on?

 Author’s response:  has been edited, thank you for pointing out the mistake.